# Long-Term PM_2.5_ Exposure Is Associated with Symptoms of Acute Respiratory Infections among Children under Five Years of Age in Kenya, 2014

**DOI:** 10.3390/ijerph19052525

**Published:** 2022-02-22

**Authors:** Peter S. Larson, Leon Espira, Bailey E. Glenn, Miles C. Larson, Christopher S. Crowe, Seoyeon Jang, Marie S. O’Neill

**Affiliations:** 1Social Environment and Health Program, Survey Research Center, Institute for Social Research, University of Michigan, Ann Arbor, MI 48104, USA; 2Department of Epidemiology, School of Public Health, University of Michigan, Ann Arbor, MI 48105, USA; crowec@umich.edu (C.S.C.); marieo@umich.edu (M.S.O.); 3Center for Global Health Equity, University of Michigan, Ann Arbor, MI 48105, USA; 4Department of Biostatistics and Epidemiology, School of Public Health and Health Sciences, University of Massachusetts, Amherst, MA 01003, USA; bglenn@umass.edu; 5Washtenaw Community College, Ann Arbor, MI 48105, USA; milarson@wccnet.edu; 6Department of Environmental Health Sciences, School of Public Health, University of Michigan, Ann Arbor, MI 48105, USA; seoyeonj@umich.edu

**Keywords:** air pollution, noncommunicable respiratory disease, asthma, chronic bronchitis

## Abstract

Introduction: Short-term exposures to air pollutants such as particulate matter (PM) have been associated with increased risk for symptoms of acute respiratory infections (ARIs). Less well understood is how long-term exposures to fine PM (PM${}_{2.5}$) might increase risk of ARIs and their symptoms. This research uses georeferenced Demographic Health Survey (DHS) data from Kenya (2014) along with a remote sensing based raster of PM${}_{2.5}$ concentrations to test associations between PM${}_{2.5}$ exposure and ARI symptoms in children for up to 12 monthly lags. Methods: Predicted PM${}_{2.5}$ concentrations were extracted from raster of monthly averages for latitude/longitude locations of survey clusters. These data and other environmental and demographic data were used in a logistic regression model of ARI symptoms within a distributed lag nonlinear modeling framework (DLNM) to test lag associations of PM${}_{2.5}$ exposure with binary presence/absence of ARI symptoms in the previous two weeks. Results: Out of 7036 children under five for whom data were available, 46.8% reported ARI symptoms in the previous two weeks. Exposure to PM${}_{2.5}$ within the same month and as an average for the previous 12 months was 18.31 and 22.1 µg/m${}^{3}$, respectively, far in excess of guidelines set by the World Health Organization. One-year average PM${}_{2.5}$ exposure was higher for children who experienced ARI symptoms compared with children who did not (22.4 vs. 21.8 µg/m${}^{3},$
*p* < 0.0001.) Logistic regression models using the DLNM framework indicated that while PM exposure was not significantly associated with ARI symptoms for early lags, exposure to high concentrations of PM${}_{2.5}$ (90th percentile) was associated with elevated odds for ARI symptoms along a gradient of lag exposure time even when controlling for age, sex, types of cooking fuels, and precipitation. Conclusions: Long-term exposure to high concentrations of PM${}_{2.5}$ may increase risk for acute respiratory problems in small children. However, more work should be carried out to increase capacity to accurately measure air pollutants in emerging economies such as Kenya.

## 1. Introduction

Climate change and rapid urbanization are leading to ever-intensifying levels of exposure to aerosol pollutants in Sub-Saharan Africa (SSA) [1]. Air pollutants, such as carbon monoxide (CO), sulfuric dioxide (SO${}_{2}$), ozone (O${}_{3}$), nitrogen dioxide (NO${}_{2}$), and particulate matter (PM), exceed World Health Organization (WHO) guidelines in many areas of SSA [2]. Ambient pollutants may increase the burden of noncommunicable respiratory diseases, such as asthma, chronic bronchitis, allergic rhinitis, and chronic pulmonary obstructive disease (COPD), following patterns seen in developed countries. Indoor pollution exposures, primarily through tobacco smoking or the use of biomass cooking fuels, have been shown to impact health in low-income countries [3,4]. Biomass cooking fuels are associated with acute respiratory infections (ARIs) [5], tuberculosis [6], COPD [7], and asthma [8,9]. Less well-understood are the associations between ambient (outdoor) air exposures to pollutants, such as PM and respiratory health in SSA.

PM is the principal component of many indoor and outdoor pollution mixtures [10]. Exposure to PM with an aerodynamic diameter <2.5 microns (PM${}_{2.5}$), also known as fine PM, has major implications for health [11]. PM${}_{2.5}$ penetrates not only the lungs’ gas exchange region, but can further penetrate into the circulatory system [12]. PM${}_{2.5}$ exposure is associated with airway inflammation, decline in lung function [13], incidence and exacerbation of asthma and COPD, and increased susceptibility to infections [12]. Upon deposition on the surface of pulmonary bronchioli and alveoli, PM${}_{2.5}$ is internalized into epithelial cells and macrophages and disrupts lung function by triggering a series of processes, including apoptosis and autophagy [12]. The most visible symptoms of PM${}_{2.5}$ exposure are the result of the activation of the inflammasome and subsequent acute and chronic responses (e.g., asthma) [12]. Persistent exposure to PM${}_{2.5}$ results in a chronic inflammatory response, worsening lung tissue injury, exacerbation of respiratory disease, and can potentially result in alveolar collapse [10].

In studies from industrialized countries, exposure to PM${}_{2.5}$ has been associated with excess hospitalizations [14]. Long-term exposures to PM${}_{2.5}$ have also been associated with increased incidence of chronic bronchitis [15], childhood asthma [16,17,18], and allergic rhinitis [19]. Cohort studies in Europe have shown that long-term exposures to NO and PM are associated with rhinitis [20]. In South Korea, increased exposure to outdoor sources of PM${}_{2.5}$ components was associated with increased odds of both coughing and wheezing in asthmatic children [21]. Children living in areas where the PM${}_{2.5}$ included components associated with electronic waste recycling in China had an elevated risk of cough, compared with children living in other areas in China [22]. Lagged concentrations of PM${}_{2.5}$ exposure in Japan were associated with cough in asthmatic people, and even stronger associations were noted among those without asthma, with indications of a dose–response relationship [23]. Moderate levels of PM${}_{2.5}$ exposure were linked with increased risk of upper respiratory tract infections in Poland [24]. PM${}_{2.5}$ concentrations measured by backpack monitors in the Bronx, New York City, USA were associated with decreased lung function in schoolchildren [25].

Kenya is a rapidly expanding lower–middle-income economy in SSA. Although 70% of the population lives in rural, undeveloped areas [26], large cities—such as Nairobi—are densely populated and highly developed. Urban populations are exposed to high ambient pollution levels, mostly from motor vehicles [27]. However, the entire Kenyan population is exposed to PM${}_{2.5}$ levels greater than the 10 µg/m${}^{3}$ WHO guideline for healthy air [28,29]. A total of 19,000 annual deaths in Kenya are attributed to air pollution, with 5000 of these thought to be due to ambient air pollution [29].

The association between ambient air pollution and health in Kenya is poorly understood. A recent review on studies that report ambient air pollutant concentrations in Kenya identified 33 studies [28]. Of those that fit the selection criteria, 23 measured PM in urban areas, with only two studies linking ambient exposure to either a health outcome or exposure levels of specific population groups [27,30]. Furthermore, the bulk of studies linking air pollution to a health outcome in Kenya have been conducted in rural areas, focusing on indoor exposure and its association with health outcomes [31,32,33,34,35,36]. To our knowledge, no study has focused on ambient exposure and its association with health outcomes nationwide in Kenya.

PM${}_{2.5}$ exposure in urban areas of Kenya has been associated with both morbidity and mortality in children under five [30]. Children in urban areas with high levels of ambient pollution were at significantly higher risk of cough, compared to those in less polluted areas. There was also a non-significant, increased risk of respiratory-related deaths in highly polluted areas [30]. A major complaint of residents in informal settlements in Nairobi was cough, often diagnosed as ARI and bronchitis [37]. Children under five in informal settlements of Nairobi had a four-times higher mortality burden than the general population, in part due to pneumonia [38]. As Kenya’s motor vehicle fleet expands, vehicular emissions will continue to rise, and pollution levels for population groups in the vicinity of roads are already at dangerously high levels [27]. Kenya also has some of the highest reported black carbon (BC) levels in the world [28]. Black carbon is a major component of PM and is formed by the incomplete combustion of fossil fuels. With the Kenyan vehicle fleet having fuel economy estimates 2–3 times worse than the vehicles’ country of origin, ambient air pollution is likely to worsen in the near future [28,39]. A better understanding of the associations between health and ambient air pollution in Kenya is therefore imperative to help with the development of mitigation measures and in the strengthening of regulatory frameworks.

Short-term exposures to PM${}_{2.5}$ and other air pollutants are known to increase risk for respiratory infections in children [40,41]. Although some research has suggested links to the development of asthma and impaired lung function [42], less well known is how long-term exposures to air pollutants, such as PM${}_{2.5}$, may raise risk for acute respiratory infections in children. In this study, we used remote sensing (satellite) data estimates of PM${}_{2.5}$ to assess the association with ARI symptoms in children under five in Kenya. The aim of this study is to understand the relationship between estimated ambient pollution levels and health outcomes and gain an understanding of the impact of the duration of exposure on health outcomes. Our definition of ARI symptomatology was derived from a question in DHS VII surveys: “When (NAME) had an illness with a cough, did he/she breathe faster than usual with short rapid breaths or have difficulty breathing?”. Though the outcome variable is subjective and likely the result of an interplay between several environmental, household, and individual factors, it may be indicative of a continuum of risk for the development of chronic conditions such as asthma. Physiologically, the symptoms mentioned in the DHS questionnaire are often the result of the narrowing of airway from the larynx to the bronchi [43]. In small children, these symptoms suggest bronchiolitis, frequently caused by respiratory syncytial virus (RSV) [43]. Given the lack of clinical diagnosis of ARI, we elected to define the outcome as “symptoms of ARI”. We use this approach to ensure comparability with other literature on ARIs in SSA using DHS data [44,45,46]. We also add to a growing number of studies using remote sensing data to understand the impact of air pollution on human health [47,48,49,50,51].

Using cluster-based survey data with predictive, satellite-based rasters of predicted PM${}_{2.5}$, this research tests three main hypotheses about exposure to PM${}_{2.5}$. First, children who experience ARI symptoms will be exposed to higher levels of PM${}_{2.5}$ than children who did not experience ARI symptoms. Second, ARI symptoms will be associated with factors associated with indoor and outdoor air pollution; including household-level factors, such as indoor cooking and smoking; and environmental factors, such as population density, living in an urban vs. rural environment, and others. Third, the associations of PM${}_{2.5}$ and ARI symptoms will vary by exposure intensity and timing, even when controlling for demographic, household, and environmental factors.

## 2. Materials and Methods

### 2.1. Individual Data

Sponsored by the United States Agency for International Development (USAID), the Demographic and Health Surveys (DHS) Program regularly collects data on population, health, HIV, and nutrition in over 90 countries. These data are deidentified and made freely available to the public and to researchers to conduct demographic and public health analyses. This research uses the Kenya Demographic and Health and Multiple Indicator Cluster Survey (EDS-MICS) of 2014 (DHS VII). The EDS-MICS is designed to collect data on household characteristics, marriage, fertility, infant and child mortality, maternal health, child health, nutrition, malaria, HIV, adult mortality, and characteristics of survey respondents such as occupation, chronic disease, and education level [52]. The EDS-MICS survey conducted sampling using a two-stage stratified, randomly drawn, national sample based on census, district, and urban vs. rural. To protect individual privacy, locations of homes are assigned randomly to clusters within the survey unit [53]. According to documentation on the DHS website, urban clusters contain a minimum of 0 and a maximum of 2 km of positional error, while rural clusters contain a minimum of 0 and a maximum of 5 km of positional error, with a further 1% of the rural clusters displaced a minimum of 0 and a maximum of 10 km.

In standard DHS surveys, all women 15–49 years of age are eligible to participate in the survey. These women’s children from 0–5 years of age are then eligible for further data collection, including specific questions on the health and nutritional status of the child and measurements of biomarkers. Relevant to this research, mothers are asked questions regarding the respiratory health of the child, specifically asking: “When (NAME) had an illness with a cough, did he/she breathe faster than usual with short, rapid breaths or have difficulty breathing?” Children whose mothers responded that children had these set of symptoms were classified as having symptoms of ARI.

### 2.2. Air Pollution Data

Estimates of monthly PM${}_{2.5}$ concentrations were obtained from the Atmospheric Composition Analysis Group (https://sites.wustl.edu/acag/, accessed on 15 December 2021). Estimates are determined from aerosol optical depth (AOD) using a physically-based relationship between AOD and PM${}_{2.5}$. Daily AOD estimates are obtained from the Moderate Resolution Imaging Spectroradiometer (MODIS) and Multiangle Imaging Spectroradiometer (MISR) satellite instruments, and coincident aerosol vertical profiles from the GEOS-Chem global chemical transport model, and transformed into grids of size 0.01 degrees square (approximately 1 km${}^{2}$ at the equator). These values are then calibrated to on-the-ground observations, using using a geographically weighted regression (GWR). The observations are then averaged to obtain monthly means. Full descriptions of the methodology used to produce the PM${}_{2.5}$ rasters are available in the scholarly literature [54,55].

Since PM${}_{2.5}$ concentrations are known to be attenuated by rainfall [56,57], precipitation data was extracted from the Climate Hazards Group InfraRed Precipitation with Station data (CHIRPS), a 35+ year quasi-global rainfall dataset [58]. Solar activity has been shown to have an impact on airborne PM${}_{2.5}$ concentrations [59]. To account for possible correlations between PM${}_{2.5}$ and sunlight, a raster for average annual sunlight as measured by Global Horizontal Irradiance (GHI) was obtained from the World Bank’s Global Solar Atlas version 2.0 [60]. Elevation can also impact PM${}_{2.5}$ concentrations due to lower air pressure [61]. Digital elevation model (DEM)-based raster data were obtained from DIVA-GIS [62]; see Figure 1. All raster data were extracted at the latitude/longitude point of the survey cluster associated with each respondent’s household. PM${}_{2.5}$ and precipitation data were extracted at each cluster point for the month of and for each of the preceding 12 months to the survey interview. To roughly assess possible confounding of remoteness and/or aridity with PM${}_{2.5}$ exposure and ARI symptoms, we calculated the Euclidean distance from the survey cluster to the nearest road and river using freely available line shapefiles [62].

### 2.3. Statistical and Analytic Methods

First, we provide descriptive statistics for the sample, including variables for ARI status, sex, age, and household wealth along with variables for indoor smoking and types of cooking fuel to account for possible indoor air pollution exposures. We also include urban and rural clusters as designated by the DHS survey. Next, we describe all exposure variables including PM${}_{2.5}$, precipitation, GHI, population, elevation, and distance to roads, rivers, and lakes. To assess possible correlations between environmental variables and PM${}_{2.5}$ levels, we produce a matrix of Pearson coefficients.

To test our first hypotheses that children with ARI symptoms will be exposed to higher levels of PM${}_{2.5}$ and that ARI symptoms will be associated with other household and environmental factors, we produce univariate logistic regression models for each variable considered. To account for possible within-cluster correlation of subjects, we include a random effect for survey cluster. Next, we use those variables to create a full model of ARI symptoms, including all variables in a single model. From there, we use a backward selection procedure to generate a best model based on Akaike’s information criterion (AIC). To test the hypothesis that ARI symptom risk will vary with PM${}_{2.5}$ by exposure intensity and timing, we use logistic regression models in a distributed lag nonlinear modeling (DLNM) framework. DLNMs offer a means to assess risk as a function of both timing and intensity of exposure. DLNMs are based on the definition of a cross-basis, which comprises two functions which describe exposure responses and the lag structure [63] and have seen increased application in health-related research [64,65,66,67,68,69], including heat related outcomes [70] and heat-related hospital admissions [71]. We created two logistic regression models using the DLNM. First, we created a model including only the crossbasis of PM${}_{2.5}$ as a predictor. Next, we used the demographic, household, and environmental variables from the optimal model chosen above to assess associations of lag exposure of PM${}_{2.5}$ with ARI symptoms.

## 3. Results

### 3.1. Sample Characteristics

In total, there were 20,964 children 0–5 years of age included in the dataset. Among these children, there were 7036 responses to the question “Did the child experience cough followed by short, rapid breath in the past two weeks?”. A total of 3526 (50.1%) of the children whose caregivers responded to the question were male. The average age of children was just slightly under two years. Nearly a third of children were from homes in the lowest SES category. A total of 93% of homes used solid fuels such as wood and charcoal for cooking. There were only 3350 responses to the question on smoking in the home, among which it was reported that 85.8% of children lived in homes where no one smokes. More than two thirds (68.2%) of children were from rural survey clusters. Demographic results are presented in the first column of Table 1. Survey clusters were spread in all areas of Kenya, including the urbanized Nairobi area; see Figure 2.

### 3.2. Environmental Measures

Latitude and longitude coordinates of survey clusters were available for 6994 (99.4%) of the children in the dataset. Mean PM${}_{2.5}$ exposures were 18.31, 22.94, and 22.1 µg/m${}^{3}$ for the month the child’s caregiver was surveyed, the month of the calendar year previous to the survey, and as a mean of the 12 calendar months prior to and including the survey month, respectively. The lowest exposures were 2, 2.3, and 10.75 µg/m${}^{3}$ for the month of the survey, the same month in the year previous, and as a mean for the year, respectively. The maximum exposures were 46.8, 66.8, and 34.49 µg/m${}^{3}$ for the month of the survey, the same month in the year previous, and as a mean for the year, respectively. Population densities ranged from extremely sparsely populated northern areas to densely populated areas of Nairobi and its environs. The distance of clusters to the nearest river or road ranged from 3.5 km to more than 25 km away in the most remote areas. Average monthly precipitation was nearly 95 mm for all time periods presented, but ranged from almost no precipitation to 334 mm in a single month and 194 mm as an average for the year. See Table 2 for full results.

PM${}_{2.5}$ measures were highly correlated with precipitation. The Pearson correlation between yearly averages of PM${}_{2.5}$ and yearly average of precipitation was 0.70. GHI was correlated weakly with same month PM${}_{2.5}$ exposures (r = 0.28). Elevation was also weakly correlated with PM${}_{2.5}$ exposures (r = 0.4). See Table 3 for the full correlation matrix.

### 3.3. Univariate Associations of Demographic and Environmental Variables with ARI Symptoms

Children who reported ARI symptoms had significantly higher exposure to PM${}_{2.5}$ in the survey month (18.9 vs. 17.8 µg/m${}^{3}$, *p* < 0.001), in the same month in the previous year (23.3 vs. 22.6 µg/m${}^{3}$, *p* = 0.036), and as an average over 12 months (22.4 vs. 21.8 µg/m${}^{3}$, *p* < 0.001). The odds of experiencing ARI symptoms were higher for males than females (odds ratio (OR) 1.13 95% confidence interval (CI) [1.04; 1.28], *p* = 0.007). Children who reported ARI symptoms were slightly younger than those who did not (1.91 years vs. 2.03 years, *p* = 0.008). Among SES groups, the odds of having ARI symptoms compared with the lowest SES group were only significantly different for households in the highest SES category (OR 0.78 95% CI [0.64; 0.94], *p* = 0.010). Although data on smoking in the home were only available for a subset of the children’s caregivers surveyed, we found no association between smoking in the home and ARI symptoms (OR 0.99 95% CI [0.82; 1.21] *p* = 0.949). Use of gas-based cooking fuels was protective against ARI symptoms compared with using biomass fuels (OR 0.70 95% CI [0.56; 0.88], *p* = 0.002). We found no association of other types of cooking fuels and ARI symptoms, but the number of responses was too small to perform a reliable statistical test. Children in urban clusters were less likely to have ARI symptoms than children in rural clusters (OR 0.87 95% CI [0.75; 0.96], *p* = 0.039). We found no association of population density and distance from the survey cluster to the nearest road or river with ARI symptoms. However, elevation, GHI, and precipitation were significantly associated with ARI symptoms. See Table 4 for full results.

### 3.4. Multivariate Model of ARI Symptoms

Very few households reported using types of cooking fuel other than gas or biomass fuels. We collapsed “electric”, “no food cooked in the house”, and “other” into a single “other” category. To maximize the dataset, we imputed missing values, imputing the mean value for numerical variables and a randomly selected category for categorical variables. We left out the variable on indoor smoking due to a large amount of missing data and since its omission did not impact the results (not shown). During our exploration, we found that since the survey cluster was the point used to extract the exposure variables, the mixed models invariably failed to converge. Given this problem, our finding that the parameter estimates from the bivariate models did not change significantly for nearly all of the variables (see Table 4) and since caterpillar plots of the random effect of survey cluster were not suggestive that a random effect was needed, we opted for fixed effects models for the analysis moving forward. We created a “full model” which included all available variables. From there, we used a backwards selection procedure to select a “best model” based on AIC. Results are shown in Table 5. The “best model” included variables for PM${}_{2.5}$ exposures (12 months previous and one year average), age, sex, type of cooking fuel, distance to lake, elevation, and the one-year average of precipitation.

### 3.5. Lag Associations of PM${}_{2.5}$ Exposure and ARI Symptoms

For the crossbasis of PM${}_{2.5}$ and precipitation, we used natural cubic splines for both the exposure and lag space. Degrees of freedom for each were set by testing multiple parameter combinations in a model including only the crossbasis, settling on the combination which yielded the lowest AIC. The parameter setting chosen was three degrees of freedom for both lag and exposure splines. We created two logistic regression models to test the association of PM${}_{2.5}$ and ARI symptoms. The first model included only the crossbasis for PM${}_{2.5}$ exposures up to lag 12 months. For the second model, we included all of the terms from the “best” multivariate model from before, but instead of including a static term for precipitation, we included the crossbasis for precipitation.

Figure 3 shows the odds of having ARI symptoms across the exposure range of PM${}_{2.5}$ and during the 12 months preceding the survey interview. For both models, we found similar patterns of exposure intensity and ARI symptoms across the lag space, though the the odds of ARI were, on average, higher in the model including demographic, household, and environmental variables. When examining the “slices” of exposure intensity with specific lags, we found that in the first model, there was a protective range of low exposures to PM${}_{2.5}$ during short lags. However, following lag 4, increased exposure to PM${}_{2.5}$ was positively associated with the odds of ARI symptoms increased along the range of exposures of PM${}_{2.5}$ and over the lag space, even when controlling for lag exposures of precipitation. Figure 4 shows the odds of having ARI symptoms across the range of PM${}_{2.5}$ exposures for lags 0, 1, 4, 6, 10, and 12. The relationship of PM${}_{2.5}$ and ARI symptoms was insignificant at higher exposure levels for earlier lags, although ARI symptoms were negatively associated with lower exposures. However, as lags increased, the association of exposure with ARI symptoms fell into patterns resembling a dose–response relationship with increasing exposures associated with increased odds of ARI symptoms. When accounting for age, sex, types of cooking fuels used, distance to lake, and lag exposures to precipitation, we found that the relationship of exposure to PM${}_{2.5}$ and ARI symptoms disappeared at short lags but fell into the same positive association for lag 4 and after that were seen in the models using only the single crossbasis for PM${}_{2.5}$; see Figure 5.

## 4. Discussion

Among this representative sample of children under age five in Kenya during 2014, our analyses suggest that longer-term exposure to PM${}_{2.5}$ is higher among children who experienced ARI symptoms in the prior two weeks than among children who did not experience ARI symptoms in the same period. This result supports our first hypothesis that PM exposure will vary between these two groups. Relevant to our second hypothesis, we have shown that ARI symptoms are associated with demographic, household, and environmental variables. For example, we have also shown that ARI symptoms in this sample were associated with the choice of cooking fuel. Households that use gas-based fuels were significantly less likely to report childhood ARI symptoms, which confirmed part of our second hypothesis. This result agrees with a wide body of literature indicating that the risk of respiratory illness in Africa is high in homes which rely on the burning of biomass for cooking and/or heat [72]. On the other hand, we did not see an association of indoor smoking with ARI symptoms. This could partly be explained by the incomplete nature of the data. The lack of an association might also be explained by the fact that smoking prevalence and frequency are relatively low in Kenya, particularly among female caregivers [73].

Regarding our third hypothesis, that ARI will be determined by PM${}_{2.5}$ exposure intensity and timing, we found evidence to suggest that long-term exposure to PM${}_{2.5}$ in children increases the odds of ARI symptoms. Specifically, we found that exposures as far as one year prior were associated with higher odds of ARI symptoms and that odds were highest among the most intense exposures. This result held even when accounting for sunlight and seasonal exposures to precipitation. This association might suggest an increased biological susceptibility to infection in children who have been exposed continuously to high concentrations of PM${}_{2.5}$ in the long term.

PM${}_{2.5}$ has been associated with an increased susceptibility to bacterial infections [74]. The first mechanism by which PM${}_{2.5}$ exposure may increase susceptibility to infection is by the promotion of bacterial adhesion to epithelial cells by the upregulation of the expression of the intercellular adhesion molecule-1 (ICAM-1, a glycoprotein on the cell surface) [74,75]. Increased pathogen adherence coupled with an impairment of the bronchial mucocilary system would result in decreased bacterial clearance, allowing pathogen buildup [74]. Another possible mechanism is the impact of PM${}_{2.5}$ exposure on the respiratory microbiome. In healthy individuals, the lower respiratory tract is typically sterile, while the upper respiratory tract has a bacterial flora that is part of the host’s natural defenses [74]. The normal flora of the upper respiratory tract provides a biological barrier against foreign matter and pathogens by a physical-space-occupying effect, nutritional competition, and the secretion of bactericidal substances [74,76,77,78]. PM${}_{2.5}$ exposure in rats has been shown to cause a decrease in indigenous flora and increase the abundance of potential pathogens, increasing susceptibility to respiratory infections [79]. The demonstrated lag that we and others have found between PM${}_{2.5}$ exposure and the onset of symptoms could be indicative of the period of time necessary for PM${}_{2.5}$ exposure to result in a buildup of pathogens and for a chronic immune response to occur. Furthermore, in infants and young children, these mechanisms are occurring against the background of the maturation of both the respiratory and immune systems. These mechanisms could have immediate and long-term effects in both later childhood and in adulthood.

Associations of long-term exposures and respiratory infections are less well understood, particularly in children. A 2013 review of particulate air pollution and acute respiratory infections [80] identified a handful of studies. One study suggested that associations between longer pollutant exposures (averaging periods of 45 days) and childhood bronchitis were stronger than associations with short-term exposures [81]. Chronic exposure to PM${}_{2.5}$ has also been associated with increased risk for infant bronchiolitis compared with short-term exposures [82]. It has also been demonstrated that increased and chronic exposure to PM${}_{2.5}$ from nearby traffic sources was significantly associated with increased odds of serious colds in children and was weakly associated with ARI symptoms [83]. In China, high concentrations of coarse PM (10–2.5) were a strong predictor of district-specific prevalence for respiratory health problems including wheezing and cough [84]. Children with long-term exposure to air pollutants in industrial Polish cities also had impaired lung development and reduced lung function compared with children from cleaner areas [13]. All the above findings coupled with our results reinforce that the effects of long-term exposures need to be better understood, particularly the association between exposures and different outcome variables such as ARIs.

The interpretation of associations should also consider that our outcome variable is a subjective set of symptoms and is not doctor-confirmed. As mentioned previously, this symptomatology is often the result of RSV infection. There have been documented associations between PM${}_{2.5}$ levels and RSV infections, with a study from Poland reporting positive associations between PM${}_{2.5}$ and RSV hospitalizations, while a study from China found a significant correlation of 0.446 between PM${}_{2.5}$ levels and the RSV infection rate [85,86]. A potential mechanism for the observed associations may be the suppression of local immunity by PM${}_{2.5}$ exposure, resulting in increased susceptibility, a longer disease, and more severe disease course, further exacerbating the oxidative stress and inflammation caused by PM${}_{2.5}$ exposure [85]. The observed lag effects between exposure and symptom presentation may be reflective of the interplay between RSV infection, ambient pollution, and the development of chronic phenotypes.

ARI phenotypes in childhood have also been used to predict lung function later in life given that repeated bronchiolitis may progress to asthma [87,88,89]. ARI phenotypes may also be indicative of the irritant nature of chemical pollutants on the immature respiratory system leading to both reversible and irreversible bronchial outcomes [87]. Furthermore, the duration of exposure to irritants may determine the progression of ARI phenotypes, with different patterns of associations between air pollution and ARI outcomes in children [90]. For example, it was found that children exposed to high levels of traffic-associated pollutants at birth were twice as likely to experience persistent wheezing at age seven [91]. However, a longer duration of exposure to high levels of traffic-associated pollutants beginning early in life was the only time period associated with the development of asthma [91]. The duration of exposure necessary for the development for ARI phenotypes may explain our findings that significant associations between PM${}_{2.5}$ levels and ARI symptoms are only present after a minimum of 4 months of exposure, possibly indicative that a cumulative threshold of exposure is necessary for symptoms to manifest. In a recent analysis of the association between ambient air pollution and respiratory health using satellite data and DHS surveys from 31 countries, no association between short-term PM${}_{2.5}$ exposure and respiratory health was found [92]. The authors used prior-month averages and evaluated two outcomes: the presence of a cough, and acute lower respiratory infection (ALRI), defined as the presence of both a cough and wheezing [92]. The lack of short-term associations, similar to our findings, reinforces the need for both more accurate ambient pollution measures as an exposure and better pathophysiological characterization of the outcome variables.

Assessing links between air pollution and health is complicated by the difficulty of properly measuring air pollution exposures. Exposure monitors are one way of assessing ambient exposures, but these are limited by the number and placement. The gridded dataset used for this study is built on satellite-based retrieval of aerosol optical depth and output from a chemical transport model (CTM). Ground-based monitors are used to understand factors that drive large-scale bias in satellite- and CTM-based estimates. The information they provide are interpreted in a way that can be applied over a large area. In the case of Kenya, only a handful of candidate monitors were available for use in the calibration for this model [93], but the predictions from the model itself are not simply locally determined. Full descriptions of the methodology used to produce the PM${}_{2.5}$ rasters are available in the literature [54,55]. A limitation, however, is the inability of the exposure raster to adequately capture localized processes which contribute to ambient PM${}_{2.5}$ levels, such as the burning of biomass fuels for cooking and/or heating. It has been suggested that grass-roots-level data collection is required to adequately assess associations between exposure and outcomes between, for example, rural and urban areas or even within urban areas themselves [94].

We found that ARI symptoms were more common in children in rural clusters than in urban clusters in the crude analyses. Several explanations are possible. First, subjects in urban and rural areas might respond differently to the question as a result of translation during the interview, differences in understanding what the question might imply, or due to differences in how subjects respond to surveys of this type. It is also possible that living conditions differ between the two contexts and the patterns of indoor exposures to pollutants might play an important role in the development of respiratory problems. Salient, nearly all (99%) households in rural areas reported using biomass cooking fuels, indicating a near daily exposure to air pollutants in and around the home. This could partly explain why the urban/rural variable was dropped from the model chosen through backward selection. More work should be undertaken to disentangle the source and effects of indoor and outdoor exposures to air pollutants.

Another major impediment to understanding the associations between air pollution and human health in SSA is a lack of national air monitor networks. Though the PM${}_{2.5}$ raster we used is calibrated based on local monitor data, the relatively low availability of such data might compromise data accuracy [28]. Further, data on the location and number of PM${}_{2.5}$ monitors used to calibrate the estimates of exposure in the gridded datasets were unavailable at the time of writing. Lacking this information, we were not able to assess the uncertainty of the exposure data used. Future studies should assess respiratory outcomes in children using personal air pollution measurement devices as has been attempted in a study in Ghana [95].

Similar to Kenya, African economies are expanding and urbanizing at an unprecedented rate. Increased access to monetary resources will mean that the number of gasoline-powered vehicles will expand, and the slow pace of adoption and the expense of renewable energy technologies might mean that Africa will depend on them moving forward. This suggests that research into exposures to PM${}_{2.5}$ and other pollutants with respiratory disease will become ever more salient in the coming years. Recent findings from the Cooking and Pneumonia Study (CAPS) in Malawi, which found that exposure to biomass fuel smoke may be less harmful than exposure to traffic-related air pollution, highlight the complexity of exposure profiles and the need for systemic mitigation measures reducing both ambient and indoor exposures [96]. As such, the development of new tools to assess ambient air pollution exposures that meet Africa’s unique contextual challenges will also be needed. With existing and new tools, researchers should work to determine the links between air pollution exposures and respiratory health given Africa’s specific set of existing health profiles.

## Figures and Tables

**Figure 1 ijerph-19-02525-f001:**
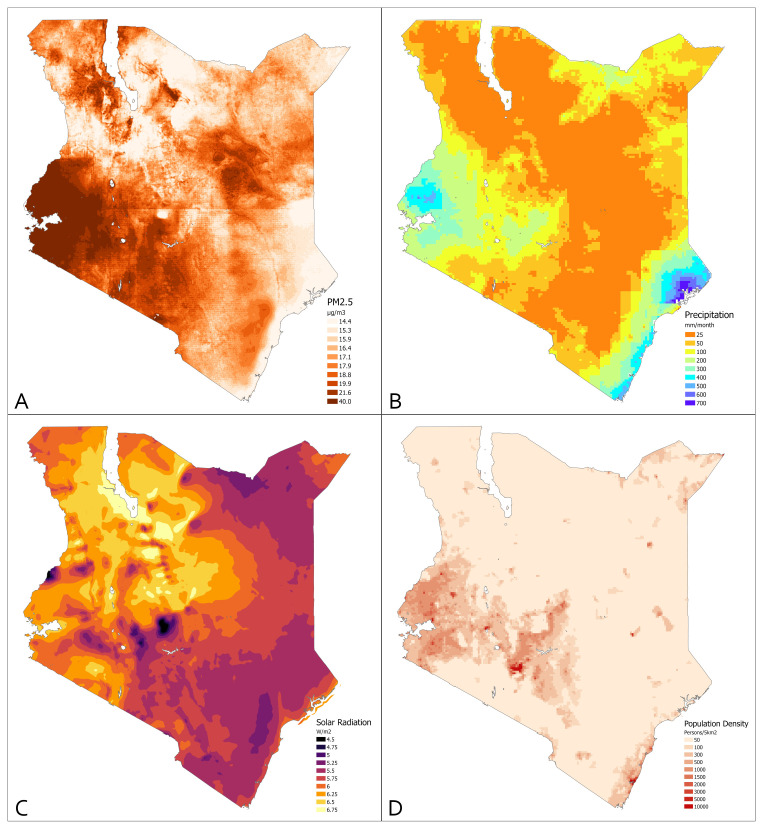
Visualizations of raster used in the analysis. (**A**) Average PM${}_{2.5}$ exposure across Kenya for the study period. (**B**) Example of CHIRPS precipitation raster model using the rainy season month of May 2014. (**C**) GHI from the World Bank Global Solar Atlas 2.0. (**D**) Gridded population of the world raster.

**Figure 2 ijerph-19-02525-f002:**
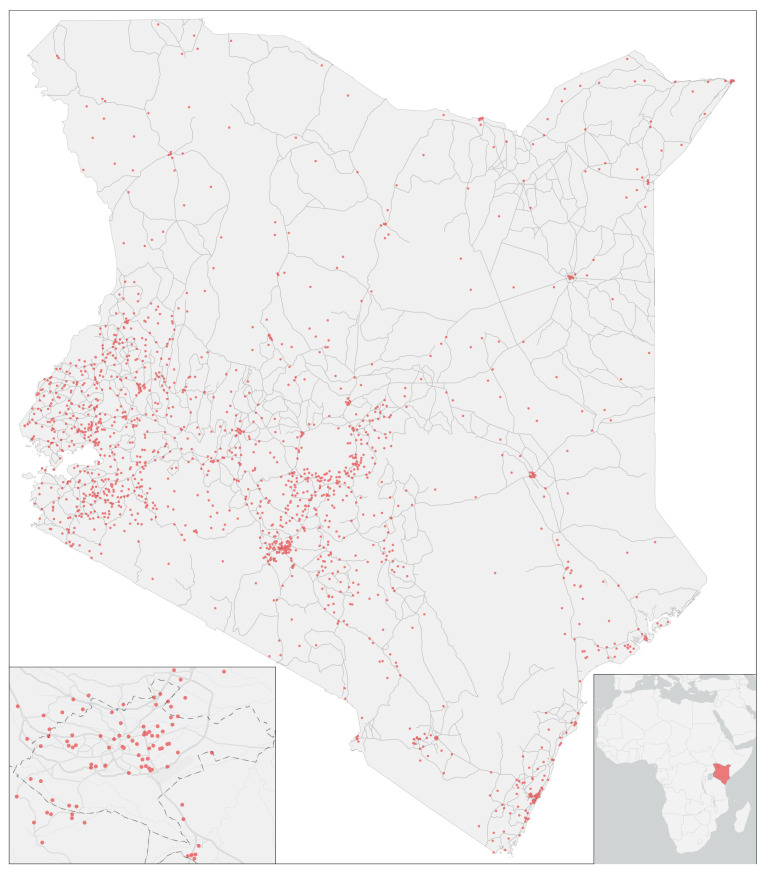
Locations of survey clusters including insets for the large urban area of Nairobi.

**Figure 3 ijerph-19-02525-f003:**
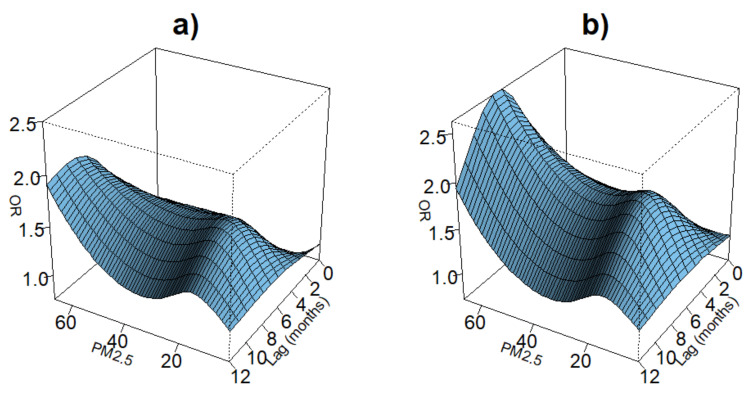
Three-dimensional plots of lag associations up to cumulative 12 months of PM${}_{2.5}$ exposure with odds ratios of symptoms of ARI. Plot (**a**) is of a model that includes only the crossbasis for PM${}_{2.5}$. Plot (**b**) is of the same model but with the additional crossbasis of precipitation and confounders for sex, age, distance to lake, and type of cooking fuel used.

**Figure 4 ijerph-19-02525-f004:**
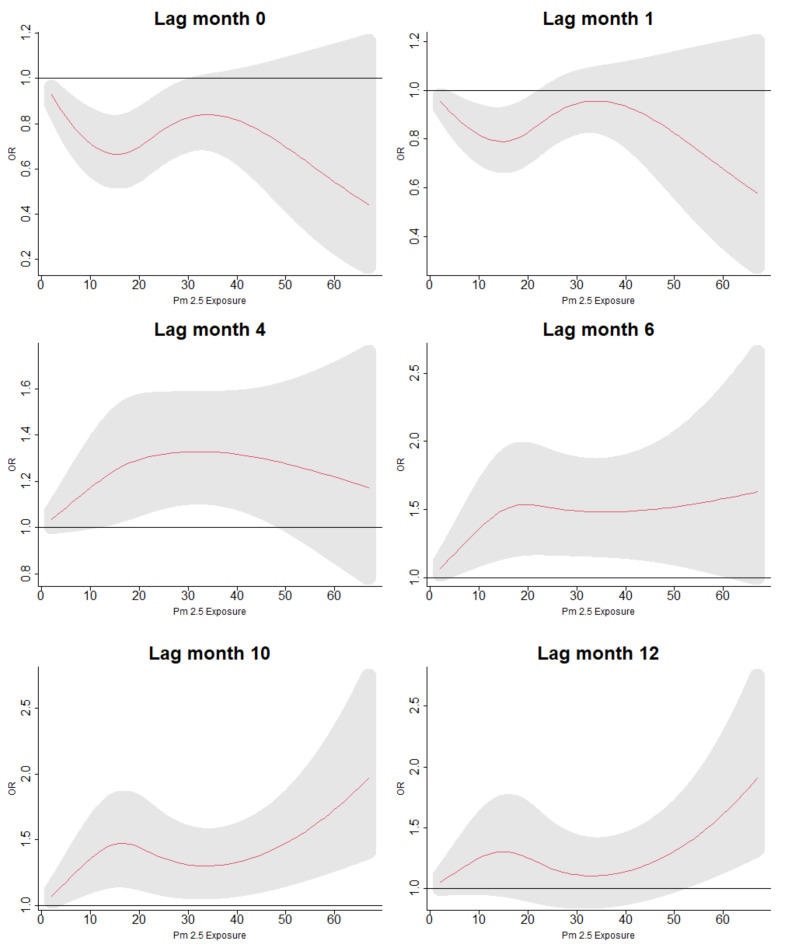
Lag-specific odds ratios of ARI with up to cumulative 12 months of exposure to PM${}_{2.5}$, including no other confounders in the model.

**Figure 5 ijerph-19-02525-f005:**
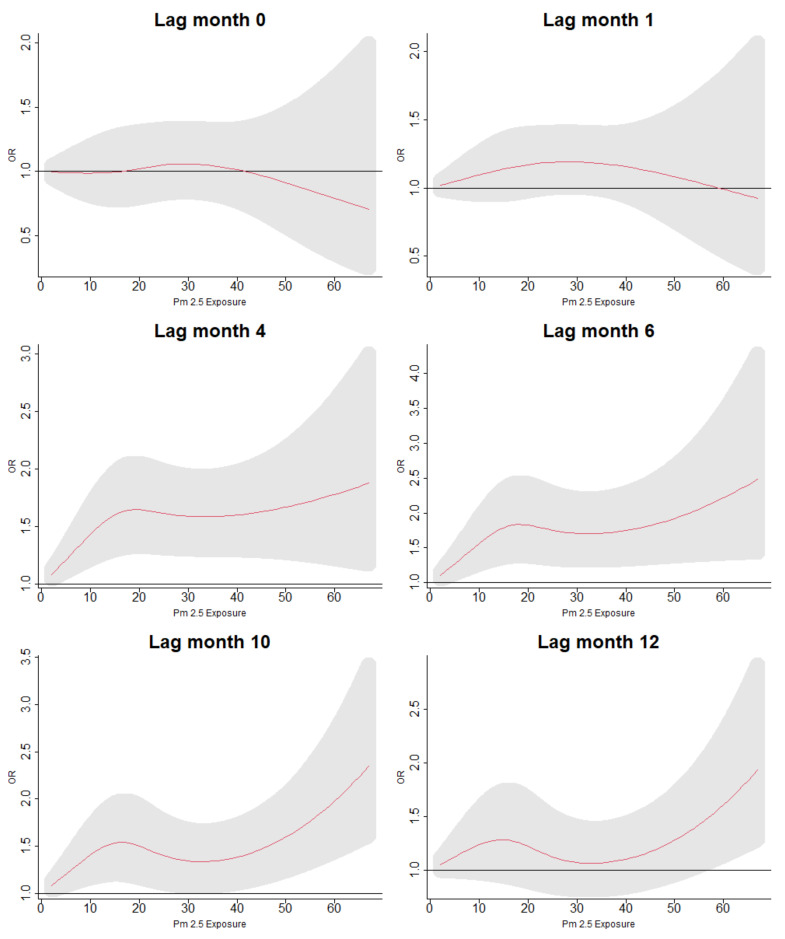
Lag-specific odds ratios of ARI with up to cumulative 12 months of exposure to PM${}_{2.5}$, including confounders for sex, age, distance from nearest lake, type of cooking fuel, and crossbassis for precipitation in the model.

**Table 1 ijerph-19-02525-t001:** Individual demographic and household characteristics for children include in the Kenya DHS VII survey from 2014 for whom information on wheezing was available.

	[ALL]	N
	N = 7036	
Wheezing:		7036
No wheezing	3744 (53.2%)	
Wheezing	3292 (46.8%)	
Sex:		7036
Female	3510 (49.9%)	
Male	3526 (50.1%)	
Current age of child (mean, std. dev.)	1.97 (1.37)	7036
Wealth index (1 = low SES, 5 = high SES):		7036
1	2174 (30.9%)	
2	1624 (23.1%)	
3	1272 (18.1%)	
4	1085 (15.4%)	
5	881 (12.5%)	
Someone ever smokes in home:		3350
No	2874 (85.8%)	
Yes	476 (14.2%)	
Type of cooking fuel used:		7035
Solid fuel	6547 (93.1%)	
Gas	470 (6.68%)	
Electricity	11 (0.16%)	
No food cooked in house	3 (0.04%)	
Other	4 (0.06%)	
Urban vs. Rural cluster:		7036
Rural	4800 (68.2%)	
Urban	2236 (31.8%)	

**Table 2 ijerph-19-02525-t002:** Summary statistics of estimated PM${}_{2.5}$ and other environmental variables in Kenyan households during 2014.

Variable	N	Mean	Std. Dev.	Min	Pctl. 25	Pctl. 75	Max
PM${}_{2.5}$ (µg/m${}^{3}$) (exposure in month of survey)	6994	18.31	9.47	2	10.7	24.3	46.8
PM${}_{2.5}$ (µg/m${}^{3}$) (12 months previous)	6994	22.94	12.94	2.3	13.4	28.4	66.8
PM${}_{2.5}$ (µg/m${}^{3}$) (one year average)	6994	22.1	4.84	10.75	18.37	25.93	34.49
Population (ppl within 5 km)	6933	1238.5	3270.49	0.15	155.48	843.01	30,644.39
Distance to road (km)	6994	3.24	3.99	0.01	0.71	4.26	41.99
Distance to river (km)	6994	3.62	3.5	0	1.11	4.79	26.97
Elevation (meters)	6972	1371.96	653.29	3	1121.25	1843	3248
Global horizontal irradiance (yearly average)	6994	5.8	0.31	4.68	5.54	6.02	6.67
Precipitation (mm) (month of survey)	6874	96.76	83.98	0	23.91	149.35	422.5
Precipitation (mm) (12 months previous)	6941	85.98	69.17	0	23.74	138.34	334.07
Precipitation (mm) (one year average)	6941	93.05	41.75	1.35	58.42	123.43	194.07

**Table 3 ijerph-19-02525-t003:** Correlation matrix of continuous environmental variables.

PM${}_{2.5}$ (month of survey)	1										
PM${}_{2.5}$ (12 months previous)	0.62	1									
PM${}_{2.5}$ (one year average)	0.77	0.69	1								
Population (1 km)	0.10	0.05	0.15	1							
Distance to road (km)	−0.12	−0.14	−0.18	−0.14	1						
Distance to river (km)	−0.09	−0.07	−0.12	0.01	0.08	1					
Elevation (meters)	0.40	0.39	0.50	0.07	−0.07	−0.14	1				
GHI (yearly average)	0.28	0.28	0.29	−0.15	0.01	−0.05	0.08	1			
Precipitation (month of survey)	0.53	0.39	0.57	0.07	−0.15	−0.09	0.26	0.27	1		
Precipitation (12 months previous)	0.61	0.36	0.49	0.03	−0.13	−0.11	0.40	0.31	0.68	1	
Precipitation (one year average)	0.70	0.55	0.72	0.12	−0.18	−0.11	0.39	0.25	0.79	0.78	1
	PM${}_{2.5}$ (month of survey)	PM${}_{2.5}$ (12 months previous)	PM${}_{2.5}$ (one year average)	Population (1 km)	Distance to road (km)	Distance to river (km)	Elevation (meters)	GHI (yearly average)	Precipitation (month of survey)	Precipitation (12 months previous)	Precipitation (one year average)

**Table 4 ijerph-19-02525-t004:** Bivariate associations of all predictors with wheezing. Means and standard deviations are presented for continuous variables. Counts and percentages are presented for categorical variables. Odds ratios and *p*-values are present for both bivariate logistic regression models with and without a random effect for survey cluster.

	No ARI	ARI	No Random Effect		Random Effect	
	N = 3744	N = 3292	OR [95% CI]	*p*	OR [95% CI]	
PM${}_{2.5}$ (month of survey)	17.80 (9.21)	18.89 (9.72)	1.012 [1.007, 1.017]	<0.001	1.013 [1.006, 1.020]	<0.001
PM${}_{2.5}$ (12 months previous)	22.63 (12.60)	23.28 (13.30)	1.004 [1.000, 1.008]	0.036	1.004 [0.999, 1.009]	0.16
PM${}_{2.5}$ (one year average)	21.83 (4.72)	22.41 (4.95)	1.025 [1.015, 1.035]	<0.001	1.026 [1.012, 1.040]	<0.001
Sex:						
Female	1923 (51.36%)	1587 (48.21%)	Ref.	Ref.		
Male	1821 (48.64%)	1705 (51.79%)	1.135 [1.033, 1.246]	0.008	1.153 [1.040, 1.278]	0.007
Current age of child	2.03 (1.37)	1.91 (1.37)	0.940 [0.909, 0.973]	<0.001	0.931 [0.897, 0.967]	<0.001
Wealth index:						
1 (low SES)	1150 (30.72%)	1024 (31.11%)				
2	836 (22.33%)	788 (23.94%)	1.059 [0.931, 1.204]	0.386	1.015 [0.872, 1.181]	0.849
3	650 (17.36%)	622 (18.89%)	1.075 [0.936, 1.234]	0.308	1.006 [0.853, 1.187]	0.94
4	589 (15.73%)	496 (15.07%)	0.946 [0.817, 1.095]	0.454	0.934 [0.784, 1.112]	0.444
5 (high SES)	519 (13.86%)	362 (11.00%)	0.783 [0.668, 0.918]	0.002	0.776 [0.640, 0.942]	0.01
Someone smokes in home:						
No smoke	1529 (85.75%)	1345 (85.83%)				
Smoke	254 (14.25%)	222 (14.17%)	0.994 [0.818, 1.207]	0.948	0.999 [0.801, 1.246]	0.993
Type of cooking fuel used:						
Solid fuel	3442 (91.96%)	3105 (94.32%)				
Gas	289 (7.72%)	181 (5.50%)	0.694 [0.573, 0.841]	<0.001	0.702 [0.559, 0.882]	0.002
Electricity	8 (0.21%)	3 (0.09%)	0.416 [0.110, 1.568]	0.195	0.405 [0.094, 1.751]	0.226
No food cooked in house	1 (0.03%)	2 (0.06%)	2.217 [0.201, 24.463]	0.516	2.906 [0.199, 42.484]	0.436
Other	3 (0.08%)	1 (0.03%)	0.370 [0.038, 3.554]	0.389	0.312 [0.026, 3.679]	0.355
Urban vs. Rural cluster:						
Rural	2501 (66.80%)	2299 (69.84%)				
Urban	1243 (33.20%)	993 (30.16%)	0.869 [0.786, 0.961]	0.006	0.866 [0.755, 0.993]	0.039
Population (1 km)	1004.97 (3211.75)	915.98 (2745.62)	1.000 [1.000, 1.000]	0.217	1.000 [1.000, 1.000]	0.33
Distance to road (km)	3.32 (4.04)	3.15 (3.92)	0.989 [0.978, 1.001]	0.078	0.992 [0.976, 1.008]	0.329
Distance to river (km)	3.63 (3.49)	3.61 (3.51)	0.998 [0.985, 1.012]	0.817	1.001 [0.982, 1.020]	0.935
Elevation (meters)	1389.66 (655.78)	1351.89 (649.97)	1.000 [1.000, 1.000]	0.016	1.000 [1.000, 1.000]	0.064
GHI (yearly average)	5.79 (0.33)	5.81 (0.30)	1.212 [1.042, 1.408]	0.012	1.282 [1.043, 1.577]	0.018
Precip (month of survey)	90.80 (80.19)	103.51 (87.61)	1.002 [1.001, 1.002]	<0.001	1.002 [1.001, 1.003]	<0.001
Precip (12 months previous)	81.45 (66.95)	91.09 (71.26)	1.002 [1.001, 1.003]	<0.001	1.002 [1.001, 1.003]	<0.001
Precip (one year average)	89.63 (39.61)	96.91 (43.72)	1.004 [1.003, 1.005]	<0.001	1.004 [1.003, 1.006]	<0.001

**Table 5 ijerph-19-02525-t005:** Full model including all covariates of interest. Final multivariate model was selected using backwards selection based on AIC. To account for missing observations, means were imputed for continuous variables. Random imputation was used for missing values in categorical variables. Poorly represented categories for cooking fuel were collapsed into a single category.

	Dependent Variable	
	ARI	
	Full Model	Best Model (AIC)
PM${}_{2.5}$ (exposure in month of survey)	1.002 *** (0.993, 1.011)	
PM${}_{2.5}$ (12 months previous)	0.994 *** (0.989, 1.000)	0.995 *** (0.990, 1.000)
PM${}_{2.5}$ (one year average)	1.023 *** (1.002, 1.043)	1.024 *** (1.006, 1.041)
Sex:		
Female	Ref.	Ref.
Male	1.131 *** (1.036, 1.226)	1.131 *** (1.036, 1.226)
Current age of child	0.942 *** (0.907, 0.977)	0.942 *** (0.907, 0.976)
Wealth index (1 = low SES, 5 = high SES):		
1	Ref.	
2	0.989 (0.848, 1.130)	
3	1.019 (0.866, 1.172)	
4	0.963 (0.798, 1.128)	
5	0.916 (0.711, 1.122)	
Type of cooking fuel used:		
Solid fuel (biomass)		
Gas	0.840 *** (0.594, 1.085)	0.809 *** (0.612, 1.006)
Other	0.587 (0.000, 1.598)	0.572 (0.000, 1.579)
Urban vs. Rural cluster:		
Rural	Ref.	
Urban	0.933 *** (0.807, 1.059)	
Population (1 km)	1.000 *** (1.000, 1.000)	
Distance to road (km)	0.993 *** (0.980, 1.007)	
Distance to lake (km)	1.003 *** (1.001, 1.004)	1.003 *** (1.001, 1.004)
Distance to river (km)	1.004 *** (0.990, 1.018)	
Elevation (meters)	1.000 *** (1.000, 1.000)	1.000 *** (1.000, 1.000)
Global horizontal irradiance (yearly average)	1.010 (0.837, 1.183)	
Precipitation (month of survey)	1.000 *** (0.999, 1.001)	
Precipitation (12 months previous)	1.000 *** (0.999, 1.001)	
Precipitation (one year average)	1.004 *** (1.002, 1.007)	1.005 *** (1.003, 1.006)
Constant	0.533 (0.000, 1.528)	0.539 *** (0.260, 0.818)
Observations	6940	6940
Log Likelihood	−4726.647	−4729.214
Akaike Inf. Crit.	9497.294	9478.428

Note: *** *p* < 0.01.

## Data Availability

Data used in this research are available from the DHS website and through other public sources.

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
