# Peer review of "Long-Term PM2.5 Exposure Is Associated with Symptoms of Acute Respiratory Infections among Children under Five Years of Age in Kenya, 2014"

_ijerph, 2022, doi:10.3390/ijerph19052525_

Round 1

Reviewer 1 Report

This is an interesting study having investigated the association between a certain type of respiratory conditions among children under the age of five and ambient levels of PM2.5 in Kenya. The authors used geo-referenced DHS-data and remote sensing based estimates of ambient PM2.5 levels.
The study is of high public health relevance as its results may also apply to other countries of Sub-Sahara Africa facing increasing levels of PM2.5 exposure from growing traffic and industrialization.

Major comments

1. I am not sure whether the condition referred to as wheezing really qualifies as being termed “wheezing”. The condition referred to is “cough followed by short rapid breathing within the two weeks preceding the survey. The prevalence of this condition is surprisingly high. Therefore, the authors should provide arguments for their use of the term wheezing for this condition.

2. From a statistical point of view, I see a few major shortcomings:

a) The authors did not adjust their analyses for geographic clustering although the data were obtained from a cluster sample. This may lead to biased precision estimates. Moreover, this does not only relate to the multivariable logistic regression models but also to the simple comparisons whose results are presented in table 4. Mixed logistic regression models or generalized estimating equations might be used for this purpose.

b) The authors based variable selection for the final multivariable model on statistical significance. However, variable selection should aim to include all potential important confounder variables, irrespective of their statistical significance. Two such variables are certainly wealth index and type of cooking fuel. Both of them showed significant differences between children with and without “wheezing” after Bonferroni adjustment. However, I would also suggest including sex and age of the child, as these variables were significantly associated with “wheezing” in the simple comparisons as well. That the authors included precipitation and global horizontal irradiance may make a lot of sense if these variables can also influence the incidence of wheezing. The variable for urban vs. rural might only be included if its p-value is small enough (e.g., < 0.2).

c) While there is a huge literature on long and short-term effects of air pollution on respiratory and other health outcomes, it is generally considered difficult to validly estimate mid-term effects of air pollution (i.e., effects from exposures having occurred more than a month and less than a year before the outcome measurement). The reason for this is the difficulty to disentangle seasonal effects unrelated to air pollution and effects of air pollution whose level varies by season.
I do not know to what extent precipitation and HDI can capture these general seasonal variations, as viral epidemics might also play a role. I therefore suggest to the authors to also run a model including the 12 month average of PM2.5 along with the lag0-average, while adding month of survey to adjust for seasonal effects unrelated to air pollution. 

Minor comments

3. The authors should be more specific on the degrees of freedom used in argvar and arglag to define the cross-bases of PM2.5 and precipitation, and on how they decided on these degrees of freedom. A useful criterion might be AIC.

4. The one-month average of PM2.5 is considerably lower than the one-year average. This suggests that the surveys were conducted mainly in seasons with lower PM2.5 levels. The authors should provide some information on the temporal distribution of surveys.

5. The authors should mention the number of monitoring stations on which remote-sensing data were calibrated and whether cross-validation of the exposure models was performed.

6. I was surprised that children from urban clusters were less likely to have “wheezing” in simple comparison.

7. I would suggest to only provide Figure 3 b, as a clear difference to Figure 3 a cannot be seen. The major differences would be at short lags, which cannot be perceived because these parts of the surface are hidden. Moreover, the legend to figure 3 b says that only precipitation was adjusted for, while GHI is mentioned as further covariate in the method section.

8. On line 292, the authors referred to their “best” model. However, they do not specify how they define this. Is it the model only containing statistically significant covariates? Anyway, this point would be obsolete if variable selection were made as suggested in 2 b).

9. The authors should add that the numbers in table 4 are means and SDs.

10. Line 187: There is an unresolved reference.

11. Line 308: The sentence seems to be incomplete

12. Line 325: “between” occurs twice

13. Line 330:  Omit “the” before “often”

14, Line 336: What is the unit of 0.134?

Author Response

1           Comments from Reviewer #1

This is an interesting study having investigated the association between a certain type of respiratory conditions among children under the age of five and ambient levels of PM2.5 in Kenya. The authors used geo-referenced DHS-data and remote sensing based estimates of ambient PM2.5 levels. The study is of high public health relevance as its results may also apply to other countries of Sub-Sahara Africa facing increasing levels of PM2.5 exposure from growing traffic and industrialization.

1.1         Major comments

  • I am not sure whether the condition referred to as wheezing really qualifies as being termed “wheezing”. The condition referred to is “cough followed by short rapid breathing within the two weeks preceding the survey. The prevalence of this condition is surprisingly high. Therefore, the authors should provide arguments for their use of the term wheezing for this

Response: Thank you for challenging us to justify our use of the term wheezing as an outcome variable. As a physiological response, wheezing is often the result of the narrowing of the airway from larynx to the small bronchi and is suggestive of bronchiolitis especially in young children. In the Kenya 2014 DHS we used Question 528 ”When (NAME) had an illness with a cough, did he/she breathe faster than usual with short, rapid breaths or have difficulty breathing” as our outcome variable that we defined as wheezing.  This outcome variable has previously been used in the literature (Harerimana et al.(2016), Seidu et al. (2019), and Mulatya and Mutuku (2020)). In all these cases the outcome was defined as either an acute respiratory infection (ARI) or an acute lower respiratory infection  (ALRI). In order to bring our definition with the literature, we have redefined our outcome as an acute respiratory infection (ARI) and no longer refer to it as wheezing. We have also included as more clear definition of our outcome variable and justify it’s use in the introduction.

We have added the following text:

”Our definition of ARI symptomatology was derived from a question in DHS VII surveys: ”When (NAME) had an illness with a cough, did he/she breathe faster than usual with short rapid breaths or have  difficulty  breathing?”.  Though  the  outcome  variable  is  subjective  and  likely  the  result  of an interplay between several environmental, household and individual factors, it may be indicative of a continuum of risk for the development of chronic conditions such as asthma. Physiologically, the symptoms mentioned in the DHS questionnaire are often the result of the narrowing of airway from the larynx to the bronchi Walker et al. [1990]. In small children, these symptoms suggest bronchiolitis, frequently caused by respiratory syncytial virus (RSV)Walker et al. [1990]. Given the lack of clinical diagnosis of ARI, We elected to define the outcome as ”symptoms of ARI.” We use this approach to ensure comparability with other literature on ARIs in SSA using  DHS dataHarerimana  et al. [2016], Seidu et al. [2019], Mulatya and Mutuku [2020].”

  • From a statistical point of view, I see a few major shortcomings:
  1. The authors did not adjust their analyses for geographic clustering although the data were obtained from a cluster This may lead to biased precision estimates. Moreover, this does not only relate to the multivariable logistic regression models but also to the simple comparisons whose results are presented in table 4. Mixed logistic regression models or generalized estimating equations might be used for this purpose.

Response: We have changed the odds ratios and p-values presented in Table 4 to reflect bivariate logistic regression models including a random effect for survey cluster , following the reviewer’s sug- gestion.  We note that results did not change appreciably; in most cases, they were exactly the same. Rather than present redundant results in tables, we decided to include only the results from the mixed models. However, when running the multivariate models, we quickly found that since the survey cluster was the point used to extract the exposure variables, the mixed models invariably failed to converge. Given this problem, our finding that the parameter estimates from the bivariate models did not change significantly and since caterpillar plots of the random effect of survey cluster were not suggestive that a random effect was needed, we opted for fixed effects models for the analysis.

  1. The authors based variable selection for the final multivariable model on statistical significance. However, variable selection should aim to include all potential important confounder variables, irrespec- tive of their statistical Two such variables are certainly wealth index and type of cooking fuel. Both of them showed significant differences between children with and without “wheezing” after Bonferroni adjustment.

Response: We have added Table 5 comparing the results of a ”full model” of all variables of interest and a ”best” model found using backward selection based on AIC. We used the ”best” model to inform our choice for models using the DLNM approach. The reviewers will note that we included both wealth index and cooking fuel in the full model. We found that the SES variable dropped out of the model.

b (cont)) However, I would also suggest including sex and age of the child, as these variables were significantly associated with “wheezing” in the simple comparisons as well. That the authors included precipitation and global horizontal irradiance may make a lot of sense if these variables can also influence the incidence of wheezing. The variable for urban vs. rural might only be included if its p-value is small enough (e.g., ¡ 0.2).

Response: The multivariable model found through backwards selection (AIC) suggested including age and sex of the child. We included these and the other variables from that model in the multivariate models of ARI symptoms that included the DLNM crossbases. Precipitation was included assuming that it would account for possible seasonality of PM2.5 exposures. There is also some evidence to suggest that precipitation could be associated with acute respiratory infections in the tropics ?.We attempted including Urban/Rural clusters, but this variable dropped out of the selected model. Taking the reviewer’s suggestion, we attempted to include urban and rural cluster in the DLNM based model, but found that the AIC increased and thus felt it prudent to leave it out of the model.

  1. While there is a huge literature on long and short-term effects of air pollution on respiratory and other health outcomes, it is generally considered difficult to validly estimate mid-term effects of air pollution (i.e., effects from exposures having occurred more than a month and less than a year before the outcome measurement). The reason for this is the difficulty to disentangle seasonal effects unrelated to air pollution and effects of air pollution whose level varies by

I do not know to what extent precipitation and HDI can capture these general seasonal variations, as viral epidemics might also play a role. I therefore suggest to the authors to also run a model including the 12 month average of PM2.5 along with the lag0-average, while adding month of survey to adjust for seasonal effects unrelated to air pollution.

Response: We thank the reviewer for their useful comment and agree that estimating mid-term effects of air pollution and health is difficult. We attempted to do as the review suggested, but found that precipitation was actually a better approximation of climatic cycles than simple month. Given that Kenya lies on the equator, seasonal variation in temperature is somewhat minimal, unlike areas in higher and lower northern and southern latitudes. Precipitation, therefore is a good approximation of seasons, but Kenya’s diversity of rainfall patterns are such that these patterns differ somewhat by region. We also note that a literature exists showing correlations of precipitation and PM2.5 so that seasonal patterns of precipitation would be sufficiently (but not perfectly) indicative of seasonal patterns in PM2.5.

1.2        Minor comments

  • The authors should be more specific on the degrees of freedom used in argvar and arglag to define the cross-bases of PM2.5 and precipitation, and on how they decided on these degrees of freedom. A useful criterion might be

Response: We have added the following text to the methods:

”For the crossbases, we used natural cubic splines for both the exposure and lag space.  Degrees of freedom for each were set by testing multiple parameter combinations in a model including only the crossbasis, settling on the combination which yielded the lowest AIC. ”

We have added the following text to the results:

”For the crossbases, degrees of freedom for the exposure and lag spaces were set at 3 and 3 respectively for both PM2.5 and precipitation.”

  • The one-month average of PM2.5 is considerably lower than the  one-year    This suggests that the surveys were conducted mainly in seasons with lower PM2.5 levels. The authors should provide some information on the temporal distribution of surveys.

Response: We have added the following text to the results:

”Interviews were conducted between April 5, 2014 and October 15, 2014.”

  • The authors should mention the number of monitoring stations on which remote-sensing data were calibrated and whether cross-validation of the exposure models was

Response: Unfortunately, these data were unavailable to us. We reached out to the team who created the PM2.5 data used in this study, but did not receive a response. We include this text in the discussion, recognizing the lack of information on the number and placement of PM2.5 monitors used for calibration and a possible limitation of the study:

”Further, data on the location and number of PM2.5 monitors used to calibrate the estimates of exposure in the gridded datasets were unavailable at the time of writing. Lacking this information, we were not able to assess the uncertainty of the exposure data used. ”

  • I was surprised that children from urban clusters were less likely to have “wheezing” in simple compar-

Response: We also found the result intriguing and added the following text to the document:

”In the crude analyses, we found that ARI symptoms were more common in children in rural clusters than in urban clusters. Several explanations are possible.  First, subjects in urban and rural areas might respond differently to the question as a result of translation during the interview, differences in understanding what the question might imply, or due to differences in how subjects respond to surveys of this type. It is also possible that living conditions differ between the two contexts and the patterns of indoor exposures to pollutants might play an important role in the development of respiratory problems. Salient, nearly all (99%) households in rural areas reported using biomass cooking fuels indicating a near daily exposure to air pollutants in and around the home. This could partly explain why the urban/rural variable was dropped from the model chosen through backward selection. More work should be done to disentangle the source and effects of indoor and outdoor exposures to air pollutants.”

  • I would suggest to only provide Figure 3 b, as a clear difference to Figure 3 a cannot be seen. The major differences would be at short lags, which cannot be perceived because these parts of the surface are Moreover, the legend to figure 3 b says that only precipitation was adjusted for, while GHI is mentioned as further covariate in the method section.

Response: We appreciate the reviewer’s suggestion. However, rather than delete the 3d plot from the multivariate model, we have rescaled the first 3d plot to match the scale of the second.  While it is true that the patterns are indeed similar (and the orientation prevents one from seeing patterns on the ”other side”), we would like to draw attention to differences between predictions from the two models. Even when including predictors including precipitation, sex and age, we see similar, though increased, patterns in the response given exposures.

  • On line 292, the authors referred to their “best” model. However, they do not specify how they define Is it the model only containing statistically significant covariates? Anyway, this point would be obsolete if variable selection were made as suggested in 2 b).

Response: We have clarified the model selection process and added a table per the suggestion from Reviewer 2. We have also included a new table demonstrating the process we used to inform our selection of models.

  • The authors should add that the numbers in table 4 are means and

Response: We have added the following text to the caption for Table 4:

”Means and standard deviations are presented for continuous variables. Counts and percentages are presented for categorical variables”

  • Line 187: There is an unresolved Response: We have corrected the reference.
  • Line 308: The sentence seems to be incomplete

Response: We have changed the text to read:

”Furthermore, in infants and young children, these mechanisms are occurring against the background of the maturation of both the respiratory and immune systems. These mechanisms could have immediate and long term effects in both later childhood and in adulthood.”

  • Line 325: “between” occurs twice

Response: We have corrected the text (also per reviewer 2.)

  • Line 330: Omit “the” before “often”

Response: We have corrected the text.

  • Line 336: What is the unit of 134?

Response: The units are hospitalizations.

For clarity, we have removed ”0.134” and changed the text to read:

”There have also been documented associations between PM2.5 levels and RSV infections, with a study from Poland reporting that a 10µg/m3 increase in PM2.5 concentration was associated with increased RSV hospitalizations, while a study from China found a significant correlation of 0.446 between PM2.5

levels and the RSV infection rate Wrotek et al. [2021], Ye et al. [2016].”

Reviewer 2 Report

The study described by the authors used georeferencing, mapping tools, and statistical approaches in order to examine the association between acute respiratory problems in young (<5 years old) children in Kenya, for which wheezing was used as a symptom marker, and fine (PM2.5) airborne particulates. The main hypotheses that were tested involve the study of associations between wheezing in children and (respectively) higher PM2.5 levels, indoor and outdoor pollution types, and variations in exposure intensity and timing.

The authors provide a good background discussion of the pathology and adverse impacts of PM2.5 exposure in the Introduction section, thereby highlighting the topic’s importance. However, given the importance of wheezing as a criterion in the study, the authors could have provided more justification for their focus on wheezing, and placed a more robust and contextual physiological description of wheezing within the Introduction section, rather than leaving it to the Discussion section where these points now reside and are explained (lines 327-365, section 4). The authors provide useful background on air pollution in Kenya, a developing east African nation for which such studies, especially ambient air pollution monitoring, have not been nearly as extensive as in wealthier nations.

Crucial to the study’s conduct are the use of several tools to measure and compare data: 1) an MICS (Multiple Indicator Cluster Survey) health data survey (in which home locations are assigned to urban/rural data clusters with a certain amount of positional error), 2) satellite spectrophotometric instruments from an atmosphere analysis group which were mapped onto geographic grids (as a measure of monthly PM2.5 concentrations), 3) precipitation, sunlight, and elevation data (all of which can impact PM2.5 levels) were mapped onto the survey cluster data as a means of further geographic sharpening of accuracy. Although the satellite PM2.5 data are calibrated to ground measurements by the company providing these and calibration is mentioned in the methods, it is unclear if the calibrations were done where corresponding PM2.5 sampling data were available, or was the PM2.5 monitoring data for calibration availalbe from the area where the health study was carried out. That distinction is important and needs to be stated clearly in detail with specific definition the calibration data in the methods section. The point is discussed superficially in the discussion (lines 371-374), and the sentence there says that the calibration is based on local data and hints that these data might be compromised. After better defining the calibration in the methods, the authors can then clearly state in the discussion the extent to which this is a limitation of the study.

Logistic regression models, either univariate or distributed lag non-linear, were used to test the various hypotheses. The described approaches are reasonable.

How data were obtained with regard to wheezing in the study children is described in lines 136-143 of section 2.1. Since wheezing is a major focus, this is a critical aspect of the study. Essentially, mothers in the aforementioned survey were provided a general descriptor for wheezing and then asked if their children aged 0-5 met that criterion. So it sounds like a “yes/no” data point was sought and recorded. This approach (being that it isn’t based on a medical or laboratory diagnostic) is subject to misinterpretation, recall bias, inaccurate or variable replies, and the inability to record grades of severity and factor that into the statistical analyses. These issues might be included in the discussion as a potential limitation of the study without impacting the value of this paper. As written in the discussion, the authors focus upon wheeze as a measureable outcome and the various pathophysiological mechanisms leading to wheeze as an outcome in epidemiological studies.

The authors found several statistically significant associations which are presented in the Results and Discussion sections and the graphs. Among them were that childhood wheezing was associated with long-term exposure to PM2.5, and also with the use of biomass fuels for indoor cooking. These are useful findings that help enhance the understanding of respiratory effects from airborne pollution, especially within the context of infrastructure and lifestyle practices in a developing African nation. They could also perhaps inform public health policy approaches in that country – a topic the authors briefly touch on in the final Discussion paragraph, and in lines 90-99 in the Introduction section, but perhaps they felt that a deeper discussion of this topic was beyond the scope of the paper.

An important potential complication of almost any type of epidemiological air pollution study is the issue of indoor pollution from tobacco use in the household. The authors did not find an association of this with wheezing, and they acknowledge (line 279, section 4) that this could be due to incomplete data. Of note, the rates of tobacco use in sub-Saharan countries such as Kenya are quite low compared to Asian, European, and North American societies. Could the authors have simply excluded the small proportion of patients (see Table 1) for which this was an issue, and analyzed the PM2.5 data on the remaining patients, thus removing this issue as a potential complication?

A key feature of the paper – both statistically and relating to their major conclusions – is the finding of “lag” associations/models that, among other things, show that  PM2.5 exposure is associated with wheezing particularly with longer time lags. To their credit, the authors propose a physiologic explanation (lines 303-309) for these lag effects in the Discussion section.

Finally, though the paper is generally well-written, a number of typos and grammatical errors are present which ideally should be corrected. A sampling is below.

Line 79 – “have a been” should be “have been”

Lines 218-219 - “to nearest river” should be “to the nearest river”

Line 324 - contains a superfluous “that”

Line 325 - contains a superfluous “between”

Lines 338, 342  – “maybe” should read “may be” (i.e. two words)

Author Response

1           Comments from Reviewer #2

  • The study described by the authors used georeferencing, mapping tools,  and statistical approaches in order to examine the association between acute respiratory problems in young (¡5 years old) children in Kenya, for which wheezing was used as a symptom marker, and fine (PM2.5) airborne The main hypotheses that were tested involve the study of associations between wheezing in children and (respectively) higher PM2.5 levels, indoor and outdoor pollution types, and variations in exposure intensity and timing.

We very much appreciate the reviewer having taken the time to read and provide such valuable and insightful comments.

  • The authors provide a good background discussion of the pathology and adverse impacts of PM2.5 exposure in the Introduction section, thereby highlighting the topic’s importance. However, given the importance of wheezing as a criterion in the study, the authors could have provided more justification for their focus on wheezing, and placed a more robust and contextual physiological description of wheezing within the Introduction section, rather than leaving it  to  the  Discussion  section  where  these  points now reside and are explained (lines 327-365, section 4). The authors provide useful background on air pollution in Kenya, a developing east African nation for which such studies, especially ambient air pollution monitoring, have not been nearly as extensive as in wealthier

Response: Thank you for pointing out that we did not properly contextualize our outcome variable. We have made several changes changes per reviewer comments. First, we no longer use the term ”wheezing” in the manuscript and instead refer to our outcome variable an acute respiratory infection (ARI). This brings our outcome definition in line with other literature using the same outcome variable. Second, we have moved our outcome definition and contextualization to the introduction to better orient the reader. The seventh paragraph of the introduction now has the following text:

”Our definition of ARI symptomatology was derived from a question in DHS VII surveys: ”When (NAME) had an illness with a cough, did he/she breathe faster than usual with short rapid breaths or have  difficulty  breathing?”.  Though  the  outcome  variable  is  subjective  and  likely  the  result  of an interplay between several environmental, household and individual factors, it may be indicative of a continuum of risk for the development of chronic conditions such as asthma. Physiologically, the symptoms mentioned in the DHS questionnaire are often the result of the narrowing of airway from the larynx to the bronchi Walker et al. [1990]. In small children, these symptoms suggest bronchiolitis, frequently caused by respiratory syncytial virus (RSV)Walker et al. [1990]. Given the lack of clinical diagnosis of ARI, We elected to define the outcome as ”symptoms of ARI.” We use this approach to ensure comparability with other literature on ARIs in SSA using  DHS dataHarerimana  et al. [2016], Seidu et al. [2019], Mulatya and Mutuku [2020].”

  • Crucial to the study’s conduct are the use of several tools to measure and compare data: 1) an MICS (Multiple Indicator Cluster Survey) health data survey (in which home locations are assigned to ur- ban/rural data clusters with a certain amount of positional error), 2) satellite spectrophotometric in- struments from an atmosphere analysis group which were mapped onto geographic grids (as a measure of monthly 5 concentrations), 3) precipitation, sunlight, and elevation data (all of which can impact PM2.5 levels) were mapped onto the survey cluster data as a means of further geographic sharpening of accuracy. Although the satellite PM2.5 data are calibrated to ground measurements by the company providing these and calibration is mentioned in the methods, it is unclear if the calibrations were done where corresponding PM2.5 sampling data were available, or was the PM2.5 monitoring data for cali- bration available from the area where the health study was carried out. That distinction is important and needs to be stated clearly in detail with specific definition the calibration data in the methods section. The point is discussed superficially in the discussion (lines 371-374), and the sentence there says that the calibration is based on local data and hints that these data might be compromised. After better defining the calibration in the methods, the authors can then clearly state in the discussion the extent to which this is a limitation of the study.

Response: We have added the following text to address the reviewer’s comment:

”Assessing links between air pollution and health is complicated by the difficulty of properly measuring air pollution exposures. Exposure monitors are one way of assessing ambient exposures, but these are limited by the number and placement. The gridded dataset used for this study is built on satellite-based retrieval of aerosol optical depth and output from a chemical transport model (CTM). Ground-based monitors are used to understand factors that drive large-scale bias in satellite- and CTM-based estimates. The information they provide are interpreted in a way that can be applied over a large area. In the case of Kenya, only a handful of candidate monitors were available for use in the calibration for this model

Open AQ, but the predictions from the model itself are not simply locally determined. Full descriptions of the methodology used to produce the PM2.5 rasters are available in the literature van Donkelaar et al. [2021, 2010].”

  • Logistic regression models, either univariate or distributed lag non-linear, were used to test the various The described approaches are reasonable.

Response: We appreciate the reviewer’s comments and support.

  • How data were obtained with regard to wheezing in the study children is described in lines 136-143 of section 2.1. Since wheezing is a major focus,  this is a critical aspect of the study.   Essentially, mothers in the aforementioned survey were provided a general descriptor for wheezing and then asked if their children aged 0-5 met that criterion. So it sounds like a “yes/no” data point was sought and This approach (being that it isn’t based on a medical or laboratory diagnostic) is subject to misinterpretation, recall bias, inaccurate or variable replies,  and the inability to record grades of severity and factor that into the statistical analyses. These issues might be included in the discussion as a potential limitation of the study without impacting the value of this paper. As written in the discussion, the authors focus upon wheeze as a measureable outcome and the various pathophysiological mechanisms leading to wheeze as an outcome in epidemiological studies.

Response:  We have clarified the outcome variable in response to comments from Reviewer 1.   We also pointed out that the outcome was based on self-report (by proxy) and was not based on a clinical diagnosis.

  • The authors found several statistically significant associations which are presented in the Results and Discussion sections and the graphs. Among them were that childhood wheezing was associated with long-term exposure to PM2.5, and also with the use of biomass fuels for indoor cooking. These are useful findings that help enhance the understanding of respiratory effects from airborne pollution, especially within the context of infrastructure and lifestyle practices in a developing African nation. They could also perhaps inform public health policy approaches in that country – a topic the authors briefly touch on in the final Discussion paragraph, and in lines 90-99 in the Introduction section, but perhaps they felt that a deeper discussion of this topic was beyond the scope of the

Response: Thank you for the insightful comment. Though we did find significant associations between our outcome and indoor cooking, we feel that an in depth exploration of this association is beyond the scope of the current paper, especially given our main exposure variable and our our subjective outcome variable . There is also a substantial body of literature that has explored this topic. In addition,recent results from the Cooking and Pneumonia Study (CAPS) in Malawi found that exposure to biomass fuel smoke maybe less harmful than exposure to traffic-related air pollution, reinforcing that this a topic area that needs further work in order to develop appropriate mitigation strategies. We have amended the last paragraph of the discussion with following text: Recent findings from the Cooking and Pneumonia Study (CAPS) in Malawi which found that exposure to biomass fuel smoke maybe less harmful than exposure to traffic-related air pollution, highlight the complexity of exposure profiles and the need for systemic mitigation measures reduce both ambient as well as indoor exposures Rylance et al. [2020].

  • An important potential complication of almost any type of epidemiological air pollution study is the issue of indoor pollution from tobacco use in the The authors did not find an association of this with wheezing, and they acknowledge (line 279, section 4) that this could be due to incomplete data. Of note, the rates of tobacco use in sub-Saharan countries such as Kenya are quite low compared to Asian, European, and North American societies. Could the authors have simply excluded the small

proportion of patients (see Table 1) for which this was an issue, and analyzed the PM2.5 data on the remaining patients, thus removing this issue as a potential complication?

Response: We appreciate the comment. We are also curiuos as to how much smoking might contribute to the development in respiratory problems in children through exposure to second hand smoke in developing countries. As economies expand, we may see smoking prevalence increase and this might become an important issue in the future. In this dataset, however, the percentage of people who reported regular smoking in the home was low. We tested to see if there was an association between smoking and ARI symptoms and found that there was little evidence to suggest there was.  We also checked for possible bias that might result from the pattern of missingness for the smoking variable and found that the pattern of missingness was unrelated to the outcome. Finally, we attempted models excluding those for which smoking data were and were not available and found that the results did not change to a large degree.

  • A key feature of the paper – both statistically and relating to their major conclusions – is the find- ing of “lag” associations/models that, among other things, show that 5 exposure is associated with wheezing particularly with longer time lags. To their credit, the authors propose a physiologic explanation (lines 303-309) for these lag effects in the Discussion section.

Response: We appreciate the reviewer’s comments and support.

  • Finally, though the paper is generally well-written, a number of typos and grammatical errors are present which ideally should be A sampling is below.

Line 79 – “have a been” should be “have been”

Response: We have corrected the text.

Lines 218-219 - “to nearest river” should be “to the nearest river”

Response: We have corrected the text. Line 324 - contains a superfluous “that” Response: We have corrected the text. Line 325 - contains a superfluous “between” Response: We have corrected the text.

Lines 338, 342 – “maybe” should read “may be” (i.e. two words)

Response: We have corrected the text.

Round 2

Reviewer 1 Report

The authors responded to all of my comments, mostly in a satisfactory way.

However, there are a few minor points left:

  1. The authors write that they tried to adjust for geographic clustering also in multivariable analyses but that models did not converge, forcing them to use models without random effects. This is fine, in principle, if there is evidence that failing to adjust for geographic clustering does not lead to biased precision estimates (point estimates can be unbiased if geographic clustering is ignored). The authors mention that the point estimates from univariable models with and without random effects were similar. But did they also compare the confidence intervals. I would suggest that they provide a table with ORs and 95%-confidence intervals from univariable models with and without random effects. This table might be placed in the on-line supplement. If it shows that both point estimates and confidence intervals are similar with and without adjusting for geographic clustering, this will support the validity of multivariable results obtained without adjustment for clustering.

  2. line 375: Give the percentage increase in RSV-hospitalisations associated with a 10 ug/m3 increment in PM2.5 or simply write that there was a positive association between PM2.5 and RSV-hospitalisations.
  3. Figure 3: Write odds ratio of symptoms of ARI rather than odds of symptoms of ARI
  4. Table 4: I would give odds ratios for 10 ug/m3 increments in PM2.5 or include 3 decimals. In the present table, odds ratios often coincide with the upper or lower limit of the 95%-confidence interval, as a consequence of rounding to two decimals.

  5. Line 452:  “reducing” instead of “reduce”

Author Response

To the editors and the reviewer,

We appreciate the reviewers insightful comments during the first round and also appreciate the careful reading of the manuscript again. 

We have responded to each of the reviewer's comments individually to the best of our ability. 
We thank the reviewer for their precious time. 

Sincerely, 

Peter S. Larson, PhD

The authors responded to all of my comments, mostly in a satisfactory way.

However, there are a few minor points left:

The authors write that they tried to adjust for geographic clustering also in multivariable analyses but that models did not converge, forcing them to use models without random effects. This is fine, in principle, if there is evidence that failing to adjust for geographic clustering does not lead to biased precision estimates (point estimates can be unbiased if geographic clustering is ignored). The authors mention that the point estimates from univariable models with and without random effects were similar. But did they also compare the confidence intervals. I would suggest that they provide a table with ORs and 95\%-confidence intervals from univariable models with and without random effects. This table might be placed in the on-line supplement. If it shows that both point estimates and confidence intervals are similar with and without adjusting for geographic clustering, this will support the validity of multivariable results obtained without adjustment for clustering.

Response: We have updated Table 4 to include both ORs and CIs for models with and without random effects. We note that the only variable to change appreciably (in significance, not extent or direction) was PM2.5 exposure 12 months previous. The model failed to converge. .

line 375: Give the percentage increase in RSV-hospitalisations associated with a 10 ug/m3 increment in PM2.5 or simply write that there was a positive association between PM2.5 and RSV-hospitalisations.

Response: We have changed the text as suggested:

"There have been documented associations between PM$ _{2.5} $ levels and RSV infections, with a study from Poland reporting positive associations betwee PM$ _{2.5} $ and RSV hospitalizations.."

Response: We have changed the text as suggested:

Figure 3: Write odds ratio of symptoms of ARI rather than odds of symptoms of ARI

"Three dimensional plots of lag associations up to cumulative 12 months of PM$ _{2.5} $ exposure with odds ratios of symptoms of ARI."

Table 4: I would give odds ratios for 10 ug/m3 increments in PM2.5 or include 3 decimals. In the present table, odds ratios often coincide with the upper or lower limit of the 95\%-confidence interval, as a consequence of rounding to two decimals.

Response: We have updated Table 4 to include 3 decimals as the reviewer has suggested.

Line 452:  “reducing” instead of “reduce”

"...highlight the complexity of exposure profiles and the need for systemic mitigation measures reducing both ambient as well as indoor exposures.."